# SARS-CoV-2 strategically mimics proteolytic activation of human ENaC

Praveen Anand[1], Arjun Puranik[2], Murali Aravamudan[2], AJ Venkatakrishnan[2]*, Venky Soundararajan[2]*

[1]nference Labs, Bengaluru, India; [2]nference, Inc, Cambridge, United States

**Abstract** Molecular mimicry is an evolutionary strategy adopted by viruses to exploit the host cellular machinery. We report that SARS-CoV-2 has evolved a unique S1/S2 cleavage site, absent in any previous coronavirus sequenced, resulting in the striking mimicry of an identical FURIN-cleavable peptide on the human epithelial sodium channel $\alpha$-subunit (ENaC-$\alpha$). Genetic alteration of ENaC-$\alpha$ causes aldosterone dysregulation in patients, highlighting that the FURIN site is critical for activation of ENaC. Single cell RNA-seq from 66 studies shows significant overlap between expression of ENaC-$\alpha$ and the viral receptor ACE2 in cell types linked to the cardiovascular-renal-pulmonary pathophysiology of COVID-19. Triangulating this cellular characterization with cleavage signatures of 178 proteases highlights proteolytic degeneracy wired into the SARS-CoV-2 lifecycle. Evolution of SARS-CoV-2 into a global pandemic may be driven in part by its targeted mimicry of ENaC-$\alpha$, a protein critical for the homeostasis of airway surface liquid, whose misregulation is associated with respiratory conditions.

*For correspondence:
aj@nference.net (AJV);
venky@nference.net (VS)

## Introduction

The surface of SARS-CoV-2 virions is coated with the spike (S) glycoprotein, whose proteolysis is key to the infection lifecycle. After the initial interaction of the S-protein with the ACE2 receptor (*Walls et al., 2020*), host cell entry is mediated by two key proteolytic steps. The S1 subunit of the S-protein engages ACE2, and viral entry into the host cell is facilitated by proteases that catalyze S1/S2 cleavage (*Belouzard et al., 2012*; *Belouzard et al., 2009*) at Arginine-667/Serine-668 (*Figure 1a*). This is followed by S2' site cleavage that is required for fusion of viral-host cell membranes (*Hoffmann et al., 2020*; *Walls et al., 2020*).

## Results

We hypothesized that the virus may mimic host substrates to achieve proteolysis. Comparing human-infecting SARS-CoV-2 with SARS-CoV strains, as well as with candidates of zoonotic origin (Pangolin-CoV and Bat-CoV RaTG13), shows that SARS-CoV-2 has evolved a unique sequence insertion at the S1/S2 site (*Zhang et al., 2020*; *Figure 1a*). Although the S protein of SARS-CoV-2 shares high sequence identity with the S proteins of Pangolin-CoV (92%) and Bat-CoV RaTG13 (97%), the furin insertion site seems to be uniquely acquired by SARS-CoV-2. The resulting tribasic 8-mer peptide (RRARSVAS) on the SARS-CoV-2 S1/S2 site is conserved among 10,956 of 10,967 circulating strains deposited at GISAID (https://www.gisaid.org/) (*Elbe and Buckland-Merrett, 2017*), as of April 28, 2020 (*Supplementary file 1a*). This peptide is also absent in over 13,000 non-COVID-19 coronavirus S-proteins from the VIPR database (*Carrillo-Tripp et al., 2009*). Strikingly, examining over 10 million peptides (8-mers) of 20,350 canonical human proteins from UniProtKB shows that the peptide of interest (RRARSVAS) is present exclusively in human ENaC-$\alpha$, also known as SCNN1A (p-value=4E-4) (see Materials and methods). The location of this SARS-CoV-2 mimicked peptide in the

**eLife digest** Viruses hijack the cellular machinery of humans to infect their cells and multiply. The virus causing the global COVID-19 pandemic, SARS-CoV-2, is no exception. Identifying which proteins in human cells the virus co-opts is crucial for developing new ways to diagnose, prevent and treat COVID-19 infections.

SARS-CoV-2 is covered in spike-shaped proteins, which the virus uses to gain entry into cells. First, the spikes bind to a protein called ACE2, which is found on the cells that line the respiratory tract and lungs. SARS-CoV-2 then exploits enzymes called proteases to cut, or cleave, its spikes at a specific site which allows the virus to infiltrate the host cell. Proteases identify which proteins to target based on the sequence of amino acids – the building blocks of proteins – at the cleavage site. However, it remained unclear which human proteases SARS-CoV-2 co-opts and whether its cut site is similar to human proteins.

Now, Anand et al. show that the spike proteins on SARS-CoV-2 may have the same sequence of amino acids at its cut site as a human epithelial channel protein called ENaC-$\alpha$. This channel is important for maintaining the balance of salt and water in many organs including the lungs. Further analyses showed that ENaC-$\alpha$ is often found in the same types of human lung and respiratory tract cells as ACE2. This suggests that SARS-CoV-2 may use the same proteases that cut ENaC-$\alpha$ to get inside human respiratory cells.

It is possible that by hijacking the cutting mechanism for ENaC-$\alpha$, SARS-CoV-2 interferes with the balance of salt and water in the lungs of COVID-19 patients. This may help explain why the virus causes severe respiratory symptoms. However, more studies are needed to confirm that the proteases that cut ENaC-$\alpha$ also cut the spike proteins on SARS-CoV-2, and how this affects the respiratory health of COVID-19 patients.

ENaC-$\alpha$ structure is in the extracellular domain (*Noreng et al., 2018*; *Figure 1b*). This suggests that the SARS-CoV-2 may have specifically evolved to mimic a human protease substrate.

ENaC regulates sodium ion (Na+) and water homeostasis, and ENaC's expression levels are controlled by aldosterone and the associated Renin-Angiotensin-Aldosterone System (RAAS)[6]. In distal lung airways, ENaC is known to play a key role in controlling fluid reabsorption at the air–liquid interface (*Rossier and Stutts, 2009*), and similar to SARS-CoV2, ENaC-$\alpha$ also needs to be proteolytically activated for its function (*Vallet et al., 1997*). FURIN cleaves the equivalent peptide on mouse ENaC-$\alpha$ between the Arginine and Serine residues in the 4[th] and 5[th] positions respectively (RSAR|SASS) (*Hughey et al., 2004a*; *Hughey et al., 2004b*), akin to the recent report establishing FURIN cleavage at the S1/S2 site of SARS-CoV-2 (*Walls et al., 2020*; *Figure 1b*). It is conceivable that human ENaC activation may be compromised in SARS-CoV-2 infected cells, for instance by SARS-CoV-2 exploiting host FURIN for its own activation. The likely consequence would be low ENaC activity on the surface of the airways leading to compromised fluid reabsorption (*Planès et al., 2010*; *Yurdakök, 2010*), an important lung pathology in COVID-19 patients with acute respiratory distress syndrome (ARDS). Indeed, the exact mechanism of SARS-CoV-2's potential impact of ENaC activation needs to be investigated.

Although the furin-like cleavage motifs can be found in other viruses (*Coutard et al., 2020*), the exact mimicry of human ENaC-$\alpha$ cleavage site raises the specter that SARS-CoV-2 may be hijacking the protease network of ENaC-$\alpha$ for viral activation. We asked whether there is an overlap between putative SARS-CoV-2 infecting cells and ENaC-$\alpha$ expressing cells. Systematic single cell expression profiling of the ACE2 receptor and ENaC-$\alpha$ was performed across human and mouse samples comprising ~1.3 million cells (*Venkatakrishnan et al., 2020*; *Figure 1c*). Interestingly, ENaC-$\alpha$ is expressed in the nasal epithelial cells, type II alveolar cells of the lungs, tongue keratinocytes, and colon enterocytes (*Figure 1c* and *Figure 2—figure supplements 1–6*), which are all implicated in COVID-19 pathophysiology (*Shweta et al., 2020*; *Venkatakrishnan et al., 2020*). Further, ACE2 and ENaC-$\alpha$ are known to be expressed generally in the apical membranes of polarized epithelial cells (*Butterworth, 2010*; *Musante et al., 2019*). The overlap of the cell-types expressing ACE2 and ENaC-$\alpha$, and similar spatial distributions at the apical surfaces, suggest that SARS-CoV-2 may be leveraging the protease network responsible for ENaC cleavage.

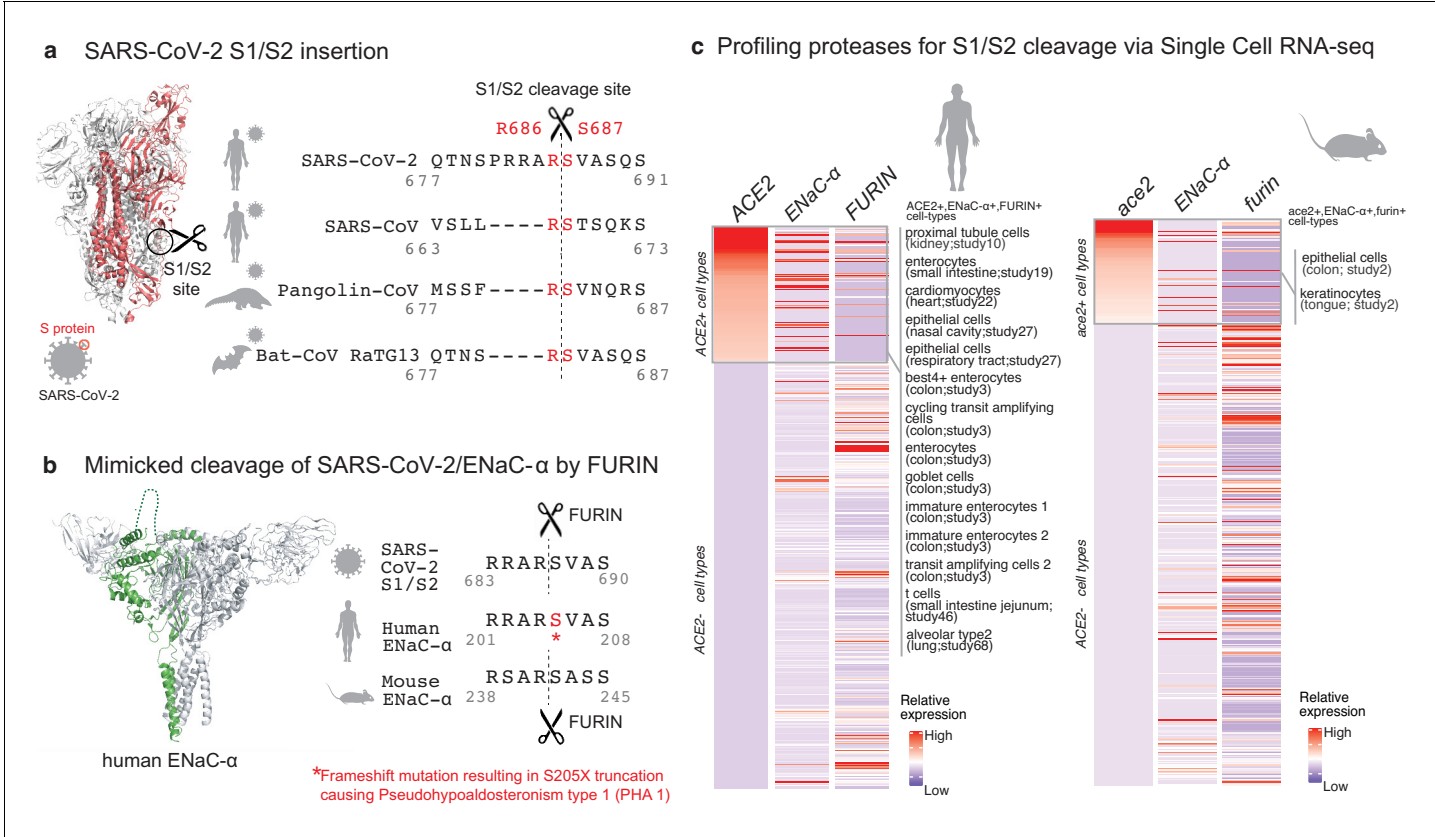

**Figure 1.** Targeted molecular mimicry by SARS-CoV-2 of human ENaC-α and profiling ACE2-FURIN-ENaC-α co-expression. (a) The cartoon representation of the S-protein homotrimer from SARS-CoV-2 is shown (PDB ID: 6VSB). One of the monomers is highlighted in red. The alignment of the S1/S2 cleavage site required for the activation of SARS-CoV-2, SARS-CoV, Pangolin-CoV, and Bat-CoV RaTG13 are shown. The four amino acid insertion evolved by SARS-CoV-2, along with the abutting cleavage site is shown in a box. (b) The cartoon representation of human ENaC protein is depicted (PDB ID: 6BQN; chain in green), highlighting the ENaC-α chain in green. The alignment on the right captures FURIN cleavage at the S1/S2 site of SARS-CoV-2, along with its striking molecular mimicry of the identical peptide from human ENaC-α protein (dotted loop in the cartoon rendering of human ENaC). The alignment further shows the equivalent 8-mer peptide of mouse ENaC-α that is also known to be cleaved by FURIN. One of the known genetic alterations on human ENaC-α is highlighted as well (*Welzel et al., 2013*). (c) The single cell transcriptomic co-expression of ACE2, ENaC-α, and FURIN is summarized. The heatmap depicts the mean relative expression of each gene across the identified cell populations. The human and mouse single cell RNA-seq are visualized independently. The cell types are ranked based on decreasing expression of ACE2. The box highlights the ACE2 positive cell types in human and mouse samples.

Beyond FURIN, which cleaves the S1/S2 site (*Walls et al., 2020*), we were intrigued by the possibility of other host proteases also being exploited by SARS-CoV-2. We created a 160-dimensional vector space (20 amino acids x eight positions on the peptide) for assessment of cleavage similarities between the 178 human proteases with biochemical validation from the MEROPS database (see Materials and methods; 0 < protease similarity metric <1) (*Rawlings et al., 2018*). This shows that FURIN (PCSK3) has overall proteolytic similarity to select PCSK family members, specifically PCSK5 (0.99), PCSK7 (0.99), PCSK6 (0.99), PCSK4 (0.98), and PCSK2 (0.94) (*Supplementary file 1b*). It is also known that the protease PLG cleaves the γ-subunit of ENaC (ENaC-γ)(*Passero et al., 2008*).

In order to extrapolate the tissue tropism of SARS-CoV-2 from the lens of the host proteolytic network, we assessed the co-expression of these proteases concomitant with the viral receptor ACE2 and ENaC-α (*Figure 2*). This analysis shows that FURIN is expressed with ACE2 and ENaC-α in the colon (immature enterocytes, transit amplifying cells) and pancreas (ductal cells, acinar cells) of human tissues, as well as tongue (keratinocytes) of mouse tissues. PCSK5 and PCSK7 are broadly expressed across multiple cell types with ACE2 and ENaC-α, making it a plausible broad-spectrum protease that may cleave the S1/S2 site. In humans, concomitant with ACE2 and ENaC-α, PCSK6 appears to be expressed in cells from the intestines, pancreas, and lungs, whereas PCSK2 is noted to be co-expressed in the pancreas (*Figure 2*). It is worth noting that the extracellular proteases

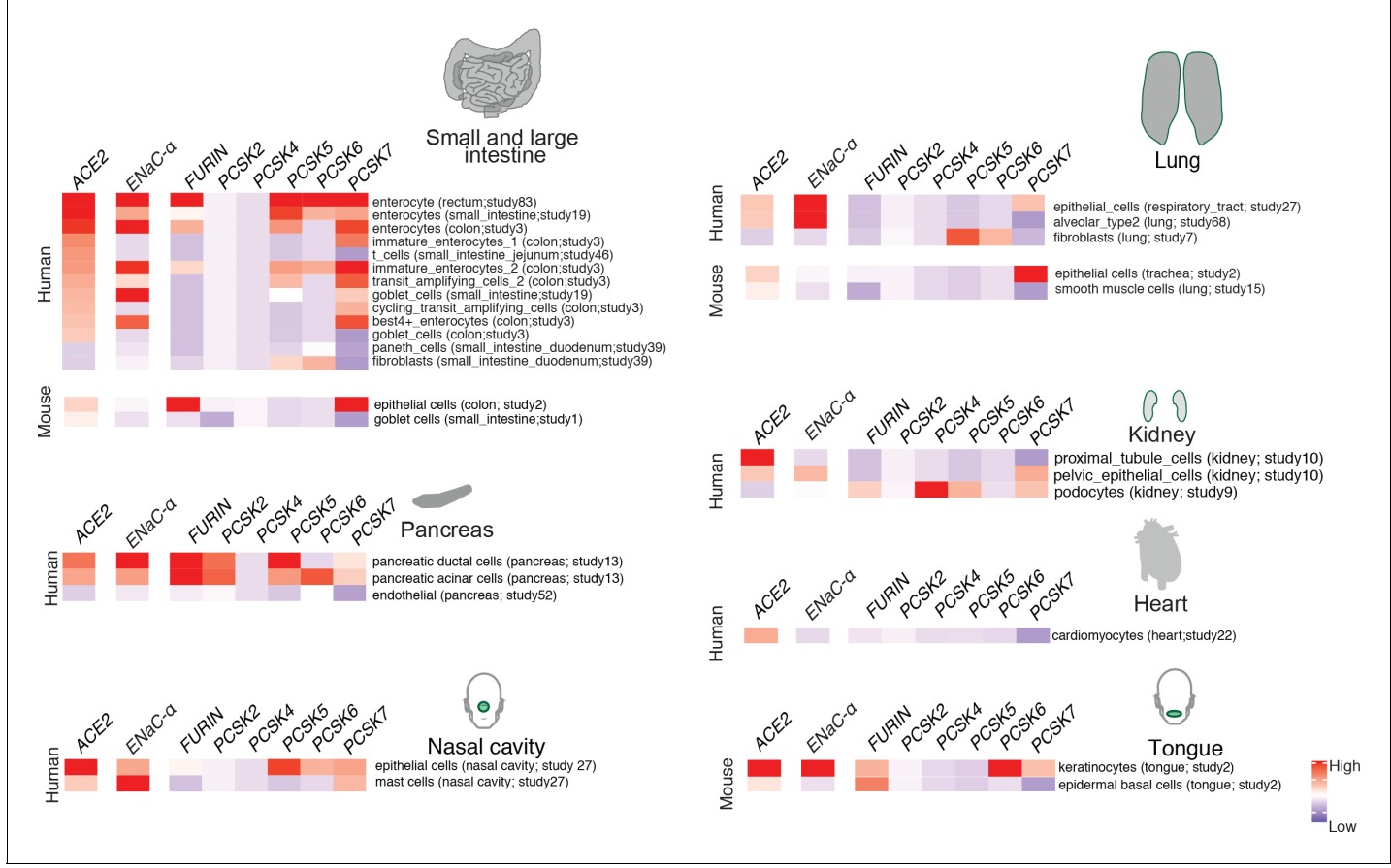

**Figure 2.** Expression profiling of identified proteases. The heatmap depicts the relative expression of ACE2 and ENaC-α along with a list of proteases that can potentially cleave the S1/S2 site. The relative expression levels are denoted on a scale of blue (low) to red (high). The rows denote proteases and columns denote cell-types.

The online version of this article includes the following figure supplement(s) for figure 2:

**Figure supplement 1.** Cardiomyocytes express ENaC-α (SCNN1A) and ACE2 (Primary data processed from Pubmed ID:31915373 and hosted on https://academia.nferx.com/).

**Figure supplement 2.** Type-II Alveolar cells of the lungs express ENaC-α (SCNN1A) and ACE2 (Primary data processed from Pubmed ID: 31892341 and hosted on https://academia.nferx.com/).

**Figure supplement 3.** Goblet cells and Ciliated cells of the nasal epithelial layer express SCNN1A (ENaC-α) and ACE2 (Primary data processed from Pubmed ID: 32327758 and hosted on https://academia.nferx.com/).

**Figure supplement 4.** Tongue keratinocytes express SCNN1A (ENaC-α) and ACE2 (Primary data processed from Pubmed ID:30283141 and hosted on https://academia.nferx.com/).

**Figure supplement 5.** Higher expression of SCNN1A was detected in 58% of the principal cells in the collecting duct 47% of the connecting tubule cells from the kidney, but ACE2 expression was not detected in these cell types.

**Figure supplement 6.** Colon enterocytes express SCNN1A (ENaC-α) and ACE2 (Primary data processed from Pubmed ID:31348891 and hosted on https://academia.nferx.com/).

need not necessarily be expressed in the same cells as ACE2 and ENaC-α. Among the PCSK family members with the potential to cleave the mimicked 8-mer peptide, it is intriguing that the same tissue can house multiple proteases and also that multiple tissues do share the same set of proteases.

## Discussion

Our findings emphasize that redundancy may be wired into the mechanisms of host proteolytic activation of SARS-CoV-2. This study should stimulate the design of experiments that confirm the working hypothesis generated by our unbiased and systematic computational analysis. The mimicry of a

cleavable host peptide central to pulmonary, renal, and cardiovascular function provides a new perspective to the evolution of SARS-CoV-2 in causing a global coronavirus pandemic.

## Materials and methods

### Alignment of coronavirus spike proteins

The complete S-protein sequence for SARS-CoV (Uniprot ID: P59594) and SARS-CoV-2 was obtained from uniprot (ftp://ftp.uniprot.org/pub/databases/uniprot/pre_release/). The sequences of Pangolin-CoV and Bat-CoV RaTG13 were obtained from the VIPR database (https://www.viprbrc.org/). Sequence alignments using Clustal-W, and comparison of SARS-CoV-2 versus other coronavirus strains were performed using JalView[17].

### Analysis of 8-mers of the human proteome

We enumerated 10,257,893 (10.26M) 8-mers from 20,350 reviewed uniprot reference sequences from human proteome (Proteome ID: UP000005640, as accessed on May 4th 2020). The previously identified SARS-CoV-2 8-mer 'RRARSVAS' was in fact found in ENaC-$\alpha$ protein (Uniprot ID: P37088; p-value $\approx 10.26M/20^8$ = 4E-4; chance of finding that particular 8-mer anywhere in the reference sequences).

### Calculating the cosine similarity metric for protease cleavage site

The position frequency matrix (PFM) of the individual proteases obtained from the MEROPS database (*Rawlings et al., 2018*) was converted to a probability weight matrix (PWM) (normalized and scaled) (*Supplementary file 1b*). Out of 178 proteases, there were 146 proteases that had specificity information available on the eight mer peptide spanning the cleavage site (±4). The 20 (amino acids) x 8 (position) matrix defined for each of the proteases were flattened into a single vector with 160 elements. We performed a cosine similarity calculation between all pairs (X,Y) of protease specificity vector. The similarity was derived as the normalized dot product of X and Y: $K(X, Y) = <X, Y> / (\|X\| * \|Y\|)$.

### Overlap of cell types expressing ENaC-$\alpha$, ACE2, and proteases from scRNA-seq datasets

We performed a systematic expression profiling of the ACE2 and ENaC-$\alpha$ across 65 published human and mouse single-cell studies comprising ~1.3 million cells using nferX Single Cell platform (*Supplementary file 1c*, https://academia.nferx.com/) (*Venkatakrishnan et al., 2020*). The ACE2 expression could be detected in 66 studies (59 studies of human samples and 7 studies of mouse samples) spanning across ~50 tissues, over 450 cell-types and ~1.05 million cells. In order to call a given cell-type to be positive for both ACE2 and a protease we applied a cutoff of 1% of the cells in the total cell-type cluster population to have a non-zero count associated with both ACE2 and the respective protease. The mean expression of the proteases, ENaC-$\alpha$ and ACE2 was derived for individual cell population within each of the studies. The cell-type information was obtained from the author annotations provided for each of the studies. The analysis was performed separately on the mouse and human datasets. For each protease, the mean expression of given cell-population (mean log[cp10k +1] counts) was Z-score normalized (to ensure the sd = 1 and mean ~0 for all the genes) to obtain relative expression profiles across all the samples. The same normalization was applied to ACE2 and ENaC-$\alpha$ and both human and mouse datasets were analyzed independently by generating heatmaps. The cell types having zero-expression values of ACE2 were also included as negative control to probe the expression of various proteases.

We performed an analysis to identify the cell types with significant overlap of ACE2 and ENaC-$\alpha$ expression. To this end, we shortlisted cell types in which ENaC-$\alpha$ is expressed in a significantly higher proportion of ACE2-expressing cells than in the overall population of cells of that sub-type. We computed the ratios of these proportions, and used a corresponding Fisher exact test to compute significance.

## Acknowledgements

The authors thank Patrick Lenehan, David Zemmour, Travis Hughes, Tyler Wagner, and Mathai Mammen for their careful review and feedback. The authors are also grateful to Ramakrishna Chilaka for the software visualization tools, and Dhruti Patwardhan, Saranya Marimuthu, Jaya Jain, Dariusz Murakowski, and Enrique Garcia-Rivera for their assistance with databases.

## Additional information

### Competing interests

Praveen Anand, Arjun Puranik, Murali Aravamudan, AJ Venkatakrishnan, Venky Soundararajan: The author is an employee of nference.

### Funding

The authors declare that there was no external funding for this work.

### Author contributions

Praveen Anand, Resources, Data curation, Software, Formal analysis, Validation, Methodology, Writing - review and editing; Arjun Puranik, Data curation, Validation; Murali Aravamudan, Validation, Methodology; AJ Venkatakrishnan, Conceptualization, Investigation, Writing - original draft, Project administration, Writing - review and editing; Venky Soundararajan, Conceptualization, Supervision, Writing - original draft, Project administration, Writing - review and editing

### Author ORCIDs

Praveen Anand (ID) https://orcid.org/0000-0002-2478-7042
AJ Venkatakrishnan (ID) https://orcid.org/0000-0003-2819-3214
Venky Soundararajan (ID) https://orcid.org/0000-0001-7434-9211

### Decision letter and Author response

Decision letter https://doi.org/10.7554/eLife.58603.sa1
Author response https://doi.org/10.7554/eLife.58603.sa2

## Additional files

### Supplementary files

• Supplementary file 1. Conservation of S1/S2 site and proteases predicted to cleave it. (**a**) SARS-CoV-2 variants in the RRARSVAS 8-mer peptide from 10,987 spike (S) protein sequences of the GISAID database. The specific variations are highlighted in **Red**. (**b**) Protease cleavage propensities for FURIN and the other proteases identified as similar from the vector space analysis conducted. Similarity (FURIN) ranges from 0 to 1. Highlighted green are amino acids occurring in greater than 10% of the cleaved substrates at that position (compiled from MEROPS). (**c**) List of single-cell studies analyzed and incorporated into the nferX resource (https://academia.nferx.com/).

• Transparent reporting form

### Data availability

All data generated or analysed during this study are included in the manuscript and supporting files.

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
