## [Decision Letter]

**Acceptance summary:**

This study entails an extensive computer analyses in silico demonstrating that the amino acid sequences of the cleavage site of protein S1/S2 of the SARS-CoV2 virus, which is essential for the penetration of the virus into the host cell, corresponds exactly to the cleavage site of ENAC-α subunit. The latter is cleaved by furin, a host protease that, in doing so, activates the Na^+^ channel. A literature analysis carried out by the authors also indicates a significant overlap between the expression of ENAC-α and ACE2, the receptor of the virus, supporting the idea that the SARS-CoV-2 virus has evolved the identity of the cleavage peptide of ENAC-α in order to facilitate its entry inside the host cell. This paper should stimulate the design of experiments aimed at confirming the working hypothesis generated by this in silico analysis.

**Decision letter after peer review:**

Thank you for submitting your article "SARS-CoV-2 strategically mimics a cleavable pulmonary renal peptide" for consideration by *eLife*. Your article has been reviewed by three peer reviewers, and the evaluation has been overseen by a Guest Reviewing Editor and Matthias Barton, MD, as the Senior Editor. The following individual involved in review of your submission has agreed to reveal their identity: Bernard Rossier (Reviewer #3).

This decision letter is to help you prepare a revised submission.

Summary:

This study entails an in silico evaluation that demonstrates, through extensive computer analyses, that the amino acid sequences of the cleavage site of protein S1/S2 of the SARS-CoV2 virus, essential for the penetration of the virus into the host cell, corresponds exactly to the cleavage site of ENAC-α subunit. The latter is cleaved by Furin, a host protease that, in doing so, activates the Na^+^ channel. A literature analysis carried out by the authors also indicates a significant overlap between the expression of ENAC-α and ACE2, the receptor of the virus, supporting the idea that the SARS-CoV-2 virus has evolved the identity of the cleavage peptide of ENAC-α in order to facilitate its entry inside the host cell.

Essential revisions:

The information that furin is the protease that cleaves the S1/S2 site of the virus is already known (Hofmann et al., Cell, 2020 , Walls et al., 2020. The previous work noted that SARS-CoV-2 contains a polybasic furin recognition site between the S1 and S2 subunits of the spike protein. In this current study, Anand et al. identified that in addition to the furin cleavage site, SARS-CoV-2 contains an 8-mer peptide that is identical to the furin cleavage site in the α subunit of ENaC located at the C-terminal end of the putative inhibitory segment. These results are striking as they seem to suggest that the virus has evolved to cleverly trick the cells by specifically mimicking an ion channel that is known to regulate salt and water in humans.

Hence, the authors should better stress the novelty of their paper, which resides in the correlation be-tween this cleavage site that is present only in SARS-CoV-2 (and not in other coronaviruses) and the ENAC-α, and by emphasizing the evolutionary aspects of their study in relation to the virus pandemics.

There is one particular section in the manuscript that authors should revise. In the third paragraph, the authors overstate the importance of the defined 8-mer peptide by citing a work that identified a premature stop codon in the middle of the 8-mer peptide that truncates the α subunit giving rise to a salt-wasting disease, PHA1. The phenotype is not due to the truncation of the 8-mer peptide but is more likely due to a lack of a formation of a functional ion channel.

Overall, the Authors should thus make an effort to put their findings in the context of the published literature on the control of airway surface liquid (including Rossier and Stutts, 2009). They are right in stating that genetic truncation at the ENaC-α cleavage site causes aldosterone dysregulation in patients. However, it would be important to stress that the furin site is critical for the activation of ENaC. If this activation site is blocked by mimicry of the unique S1/S2 cleavage site, its likely consequence would be very low ENaC activity on the surface of the airways, compromising severely fluid reabsorption, which can explain an important lung phenotype observed in COVID19. The so called "wet lung" syndrome, that can occur in premature babies because ENaC is only fully expressed at the time of birth, should be mentioned to support this contention. The PHA Type 1 case described in Welzel et al., 2013, also supports the importance of the furin site for ENaC activation.

Additional issues:

1) The authors reported in Figure 2—figure supplement 5 ENaC expression in the proximal tubule, which is wrong as ENac is expressed only in the aldosterone sensitive distal nephron (ASDN) (Distal Convoluted Tubule DCT2, Connecting Tubule CNT and Collecting Duct CD).

2) References should be better placed and selected. For example in the second paragraph, last sentence: the structure of human ENaC (Noreng et al., 2018) should be cited here and not in the third paragraph where earlier work on structure by Vallet et al., 1997 or by Orce et al., 1980 should be cited.

3) The title is too generic and should mention ENaC rather than “a cleavable pulmonary renal peptide”.

---

## [Author Response]

Essential revisions:The information that furin is the protease that cleaves the S1/S2 site of the virus is already known (Hofmann et al., Cell, 2020 , Walls et al., 2020. The previous work noted that SARS-CoV-2 contains a polybasic furin recognition site between the S1 and S2 subunits of the spike protein. In this current study, Anand et al. identified that in addition to the furin cleavage site, SARS-CoV-2 contains an 8-mer peptide that is identical to the furin cleavage site in the α subunit of ENaC located at the C-terminal end of the putative inhibitory segment. These results are striking as they seem to suggest that the virus has evolved to cleverly trick the cells by specifically mimicking an ion channel that is known to regulate salt and water in humans.Hence, the authors should better stress the novelty of their paper, which resides in the correlation be-tween this cleavage site that is present only in SARS-CoV-2 (and not in other coronaviruses) and the ENAC-α, and by emphasizing the evolutionary aspects of their study in relation to the virus pandemics.

As suggested, we have now updated the manuscript to highlight the evolutionary aspects of our study in relation to virus pandemics. Figure 1A has been updated to compare the S1/S2 cleavage sites from human infecting SARS-CoV and SARS-CoV-2 as well as with candidates of zoonotic origin (Pangolin-CoV and Bat-CoV RatG13). We have also updated the text accordingly:

“We hypothesized that the virus may mimic host substrates in order to achieve proteolysis. […] This suggests that the SARS-CoV-2 may have specifically evolved to mimic a human protease substrate.”

“Although the furin-like cleavage motifs can be found in other viruses (Coutard et al., 2020), the exact mimicry of human ENaC-ɑ cleavage site raises the specter that SARS-CoV-2 may be hijacking the protease network of ENaC-ɑ for viral activation.”

There is one particular section in the manuscript that authors should revise. In the third paragraph, the authors overstate the importance of the defined 8-mer peptide by citing a work that identified a premature stop codon in the middle of the 8-mer peptide that truncates the α subunit giving rise to a salt-wasting disease, PHA1. The phenotype is not due to the truncation of the 8-mer peptide but is more likely due to a lack of a formation of a functional ion channel.

We thank the reviewers for this feedback. We now clarify this better:

“ENaC regulates sodium ion (Na+) and water homeostasis and ENaC’s expression levels are controlled by aldosterone and the associated Renin-Angiotensin-Aldosterone System (RAAS)(Elbe and Buckland-Merrett, 2017^)^. […] The PHA1 phenotype is likely due to a lack of formation of a functional ion channel, and highlights the salience of the FURIN site for ENaC activation.”

Overall, the Authors should thus make an effort to put their findings in the context of the published literature on the control of airway surface liquid (including Rossier and Stutts, 2009). They are right in stating that genetic truncation at the ENaC-α cleavage site causes aldosterone dysregulation in patients. However, it would be important to stress that the furin site is critical for the activation of ENaC. If this activation site is blocked by mimicry of the unique S1/S2 cleavage site, its likely consequence would be very low ENaC activity on the surface of the airways, compromising severely fluid reabsorption, which can explain an important lung phenotype observed in COVID19. The so called "wet lung" syndrome, that can occur in premature babies because ENaC is only fully expressed at the time of birth, should be mentioned to support this contention. The PHA Type 1 case described in Welzel et al., 2013, also sup-ports the importance of the furin site for ENaC activation.

We thank the reviewer for this feedback. We have now revised the text accordingly.

“Among many reported loss-of-function mutations in ENaC-ɑ associated with Pseudohypoaldosteronism type 1 (PHA1), a frameshift mutation leading to a premature stop codon in Serine-205 at the 5^th^ position of the ENaC-ɑ mimicked peptide (RRAR|SVAS) is also known to cause this monogenic disease (Welzel et al., 2013). […] Indeed, the exact mechanism of SARS-CoV-2’s potential hijack of ENaC activation needs to be investigated.”

Additional issues:1) The Authors reported in Figure 2—figure supplement 5 ENaC expression in the proximal tubule, which is wrong as ENac is expressed only in the aldosterone sensitive distal nephron (ASDN) (Distal Convoluted Tubule DCT2, Connecting Tubule CNT and Collecting Duct CD).

We thank the authors for pointing this out. Figure 2—figure supplement 5 has now been updated to include the principal cells from collecting duct along with connecting tubule cells. Although both these cell-types express high levels of ENAC-α (58% and 47%), ACE2 expression was not detected in these cells. In contrast, only 2.77% of the proximal tubule cells had detectable expression of SCNN1A, but a higher percentage (8.46%) of these cells were also observed to express ACE2.

2) References should be better placed and selected. For example in the second paragraph, last sentence: the structure of human ENaC (Noreng et al., 2018) should be cited here and not in the third paragraph where earlier work on structure by Vallet et al., 1997 or by Orce et al., 1980 should be cited.

We now have updated the references as suggested.

3) The title is too generic and should mention ENaC rather than “a cleavable pulmonary renal peptide”.

We thank the reviewer for this feedback. We have now updated the title to: “SARS-CoV-2 strategically mimics proteolytic activation of human ENaC”.